# Comparative Analysis of Type 1 and Type Z Protein Phosphatases Reveals D615 as a Key Residue for Ppz1 Regulation

**DOI:** 10.3390/ijms23031327

**Published:** 2022-01-25

**Authors:** Antonio Casamayor, Diego Velázquez, Carlos Santolaria, Marcel Albacar, Morten I. Rasmussen, Peter Højrup, Joaquín Ariño

**Affiliations:** 1Institut de Biotecnologia i Biomedicina & Departament de Bioquímica i Biologia Molecular, Universitat Autònoma de Barcelona, 08193 Cerdanyola del Vallès, Spain; Antonio.Casamayor@uab.cat (A.C.); diego.velazquez91@icloud.com (D.V.); Carlos.Santolaria@uab.cat (C.S.); Marcel.Albacar@uab.cat (M.A.); 2Department of Biochemistry and Molecular Biology, University of Southern Denmark, Campusvej 55, 5230 Odense, Denmark; morras@massai.dk (M.I.R.); php@bmb.sdu.dk (P.H.)

**Keywords:** protein phosphatase, Ppz1, PP1c, sequence comparisons, cross-linking proteomics, enzyme inhibition, fungi

## Abstract

Type 1 Ser/Thr protein phosphatases are represented in all fungi by two enzymes, the ubiquitous PP1, with a conserved catalytic polypeptide (PP1c) and numerous regulatory subunits, and PPZ, with a C-terminal catalytic domain related to PP1c and a variable N-terminal extension. Current evidence indicates that, although PP1 and PPZ enzymes might share some cellular targets and regulatory subunits, their functions are quite separated, and they have individual regulation. We explored the structures of PP1c and PPZ across 57 fungal species to identify those features that (1) are distinctive among these enzymes and (2) have been preserved through evolution. PP1c enzymes are more conserved than PPZs. Still, we identified 26 residues in the PP1 and PPZ catalytic moieties that are specific for each kind of phosphatase. In some cases, these differences likely affect the distribution of charges in the surface of the protein. In many fungi, Hal3 is a specific inhibitor of the PPZ phosphatases, although the basis for the interaction of these proteins is still obscure. By in vivo co-purification of the catalytic domain of ScPpz1 and ScHal3, followed by chemical cross-linking and MS analysis, we identified a likely Hal3-interacting region in ScPpz1 characterized by two major and conserved differences, D566 and D615 in ScPpz1, which correspond to K210 and K259 in ScPP1c (Glc7). Functional analysis showed that changing D615 to K renders Ppz1 refractory to Hal3 inhibition. Since ScHal3 does not regulate Glc7 but it inhibits all fungal PPZ tested so far, this conserved D residue could be pivotal for the differential regulation of both enzymes in fungi.

## 1. Introduction

Analysis of the *Saccharomyces cerevisiae* genome allows the identification of at least 19 proteins with predicted or experimentally tested Ser/Thr protein phosphatase activity. Twelve of these can be classified within the PPP (phosphoprotein phosphatase) superfamily, which includes homologs of the broadly distributed type 1 (PP1), 2A (PP2A) and 2B (PP2B) phosphatases, accounting for most of the serine/threonine phosphatase activity in eukaryotic cells. From a structural point of view, they all share a conserved central β sandwich that coordinates two metal ions [1,2]. In *S. cerevisiae*, three different proteins represent type 1 phosphatases: Glc7, Ppz1/2, and Ppq1. *GLC7* is an essential gene that encodes a 312-residue protein, mostly comprising the canonical catalytic PPP domain. It can be considered the yeast homolog of the ubiquitous eukaryotic type 1 catalytic subunit (PP1c) and shows a high level of sequence conservation (>80%), even among distant species. Ppz and Ppq1 enzymes contain a conserved C-terminal PPP domain preceded by an N-terminal extension that is structurally unrelated. In contrast to Glc7, phosphatases Ppz1/2 and Ppq1 are found only in fungi. However, while Ppz enzymes can be found in virtually any fungal species, Ppq1 has a much more restricted distribution, as it is only found in some of the families of Saccharomycetales, such as Saccharomycetaceae, Pichiaceae or Debaryomycetaceae (but it is absent in Dipodascaceae and Trichomonascaceae), and little is known about its function and regulation. For these reasons, the Ppq protein family is not included in this study.

Glc7 plays a myriad of cellular roles, including glucose repression, translation, transcription, cation homeostasis, and cell cycle regulation. The enormous functional diversity of eukaryotic type 1 enzymes is explained by its interaction with a plethora of regulatory subunits that modulate activity, localization, or substrate selection of the catalytic subunit. The association of Glc7 with not less than 30 such proteins was reported [1,2,3]. The characterization of the structural basis for the interaction between PP1c and its regulatory subunits was carried out mostly with mammalian enzymes [4], and it involves docking motifs, that is, short sequences (about 4–8 residues) present in the regulatory subunits that, in combination, create a larger interaction surface for PP1c. About 10 known distinct PP1-docking motifs were identified in the regulatory subunits in mammals, although not all of them are found in yeast [5]. The most general and conserved motif is the “RVxF” consensus sequence, which interacts with the hydrophobic groove of PP1c. As expected for such finely regulated activity, the overexpression of Glc7 is detrimental for the cell [6,7].

In *S. cerevisiae,* Ppz proteins are represented by two paralogs, Ppz1 and Ppz2 (692 and 710 residues, respectively), although most fungi are endowed with a single Ppz-encoding gene. As mentioned above, the ScPpz1 catalytic PPP domain occupies the C-terminal half of the protein (about 340 residues), which is about 60% identical to Glc7, and it is preceded by a nearly 350-residue-long, much more divergent N-terminal region [8,9]. Ppz phosphatases were characterized in some detail in several fungi [10,11,12,13,14,15], and were, in some cases, identified as a virulence determinant [16,17,18]. As a rule, the catalytic domain appears quite conserved, whereas the N-terminal extension is highly variable or almost nonexistent in some cases. It is generally accepted that Ppz phosphatases play an important role in monovalent cation homeostasis. In *S. cerevisiae*, Ppz1 is also involved in the regulation of ubiquitin, dephosphorylation of the ubiquitin ligase adaptor Art1, and translation initiation (see [2] for references).

In contrast to Glc7, only two regulatory subunits, Hal3 and Vhs3, were determined for ScPpz1, Hal3 being the major in vivo regulator. Both proteins bind to the catalytic domain of the phosphatase and inhibit its phosphatase activity [19,20,21,22,23]. It must be noted that in *S. cerevisiae* and other related fungi, Hal3 and Vhs3 are moonlighting proteins involved in the biosynthesis of coenzyme A (CoA). This occurs by interaction with a third structurally related protein (Cab3), thus forming an atypical heterotrimeric phosphopantothenoylcysteine decarboxylase (PPCDC) enzyme [24].

The evidence suggests that the regulation of Ppz1 activity must be exceptionally effective, since even moderately higher than normal levels of the phosphatase cause a dramatic cell growth arrest [19,25]. Genetic approaches indicate that, when overexpressed, Ppz1 could be the most toxic protein in budding yeast [26], and recent evidence suggests that such toxicity derives from the alteration of multiple targets that are likely different from those of Glc7 [27,28]. An important question is how PP1 and PPZ regulation (and therefore cellular functions) can be insulated. In fact, such insulation is likely not perfect, since it is known that Ppz1 can interact with certain Glc7 subunits endowed with the RVxF interaction motif, such as Glc8 and Ypi1 [29,30]. In addition, Ppz1 has a moderate sensitivity to mammalian inhibitor-2, a well-known regulatory component for PP1c [31,32]. Although there is evidence suggesting that the N-terminal extension of Ppz phosphatases could influence their function and affect interaction with the Hal3 inhibitor [13,19,25,33,34], the fact that Hal3 mainly interacts with the catalytic C-terminal domain suggests that this region should contain specific structural determinants for regulation. Important insights were derived from the 3D structure of the catalytic domain of *C. albicans* Ppz1 (CaPpz1) [32], revealing structural determinants that could allow regulation of this enzyme by PP1c-specific regulators and an additional structured C-terminal α-helix not present in PP1c enzymes.

Most of our current knowledge on this field derives from the study of a very limited number of species (generally *S. cerevisiae* and *C. albicans*). Therefore, we considered that extension of the analysis across the fungal kingdom may provide a set of common and distinctive characteristics from which more general rules on the differential regulation of fungal PP1 and PPZ phosphatases could be derived. Such study has led to the identification of a strongly conserved residue (D615) that is most likely relevant for specific functions of Ppz1.

## 2. Results and Discussion

We selected 57 species as representatives of three main different fungi clades (Figure 1) according to a proteome-based classification [35]. The number and distribution of PP1 and Ppz proteins analyzed are shown in Table 1, and the identities of the selected species are presented in Appendix A. Most of these organisms possess one PP1 and one Ppz1, but a few exceptions are observed. Thus, several members of the *Saccharomycetaceae* family, corresponding to the post-WGD lineage (*S. cerevisiae*, *C. glabrata* and *K. lactis*) have two genes coding for PPZ proteins, and about seven very similar PPZ proteins are annotated in the Zygomycota *Rhizopus delemar* genome.

Concerning PP1, two proteins are found in a set of species included in the Taphrinomycotina subphylum that correspond to the *Schizosaccharomycetaceae* family (the fission yeast *S. pombe*, *S.*
*japonicus*, *S. cryophilus* and *S. octosporus*). In addition to the “standard” PP1, these organisms possess a “non-conventional” PP1, named Sds21, which differs in a few structural aspects from the canonical PP1 proteins. The Agaricomycotina *Calocera viscosa* and the Chytridiomycota *Gonapodya prolifera* and *Allomyces macrogynus* also possess two PP1 proteins.

### 2.1. Sequence Comparison of Catalytic Domains across Fungi Identifies Conserved Ppz-Specific Residues

Alignment of the selected 131 sequences corresponding to the catalytic domain of PP1 and PPZ proteins using Clustal Omega revealed that 141 residues are conserved in PP1, whereas only 50 are maintained in PPZ proteins (Appendix A). Forty-three of these residues are shared between both types of phosphatases. However, a closer inspection of the data clearly indicates that this relatively low conservation level (mainly affecting PPZ enzymes) is caused by the inclusion of “Monokaryotic” fungi in the analysis. Indeed, when these organisms are excluded, 171 residues (56.7%) are conserved in PP1, and 137 (41.0%) in PPZ enzymes. Of these, 104 are common to both PP1 and PPZ proteins. In addition, the phylogenetic tree generated from the above-mentioned Clustal Omega analysis revealed that the sequences of the PP1 proteins are much more homogeneous than the sequences of the catalytic region of the analyzed PPZs. In fact, the most divergent PP1 sequences correspond to the Sds21 proteins, the second type of PP1 phosphatase found specifically in fission yeasts (Figure 2). These results indicate that the structure of PP1 enzymes shows a higher level of conservation across fungi than that of PPZ enzymes.

Most differences between PP1 and PPZ catalytic domains are listed in Appendix A and can be grouped into four categories. In the first one (class A), PP1 and PPZ differ in 26 residues whose nature is invariant or largely maintained in both types of enzymes. Twenty-seven residues are virtually constant in PP1 but present a substantial variation in PPZs (class B). The third group (C) is composed of residues found to be constant in PPZ but variable in PP1 (C) and is substantially smaller (only three cases). A fourth group (D), containing five instances, corresponds to differences between both enzymes that are only found in specific clades. As can be observed in Figure 3, when models for ScGlc7 and ScPpz1 are constructed, many residues from these groups map onto the surface of both proteins. Interestingly, several of the class A differences, such as N484, T486, V488 or M604, map near the PPZ catalytic site. We considered this last group as particularly relevant to predict possible PPZ-specific functional features.

Because of the possible relevance of charged residues in the interaction with regulatory proteins, we were interested in their distribution on the surface of both types of enzymes. To this end, we identified those positions for which a change in charge could be predicted for all species (or at least for specific orders or families, Appendix A). We identified 30 such changes, of which 23 can be considered quite constant. In 14 of those, changes were from acidic in PP1 to neutral residues in PPZ (or vice versa), whereas 7 implied the substitution of a basic residue by a neutral one (or vice versa). Two remarkable modifications were PP1c K210 and K259, which were changed to D in the PPZ of all fungi evaluated (D566 and D615 in ScPpz1). These residues were mapped on the surface of models constructed for ScGlc7 and ScPpz1 (Figure 4 and see below). As it can be seen, the distribution of charges is not too different on the surface surrounding the extremely conserved catalytic site (panels A), but it is appreciable in many other areas of both molecules.

### 2.2. The PP1 Structural Features Required for Modulation by Regulatory Subunits Appear Largely Conserved in Fungi

Most regulatory subunits of mammalian PP1 possess a conserved “RVxF” motif that interacts with the hydrophobic groove of PP1c, which is fully conserved in fungal PP1 and strongly maintained in PPZ proteins (Appendix A). Such conservation most likely explains the observed interactions between ScPpz1 and diverse Glc7 regulatory subunits, such as Glc8 (the homolog of mammalian inhibitor-2), Ypi1, Gip2 or Sla1 [29,30], as well as the sensitivity of ScPpz1 and CaPpz to inhibitor-2 [32,37]. It also suggests that these regulatory interactions are maintained in most fungi, if not all. Mammalian inhibitor-2 binds to PP1c through the RVxF and the SILK motifs plus a long helix, which expands across the PP1 active site [38]. While the SILK motif is conserved in all fungal PP1 (only a conservative change of I for L in position 58 in a few species), it is not conserved in PPZ. In contrast, the long helix is strongly conserved in both PP1 and PPZ enzymes (Appendix A). Since it has been proposed that Hal3 blocks the activity of the phosphatase by interacting with this region [39], this would explain why ScHal3 was able to inhibit in vitro all PPZ tested so far.

Recent work by Choy and coworkers [40] identified the residues in PP1c required for interaction with the regulatory subunit Sds22, which lacks the RVxF motif and is composed of a short unstructured N-terminal region followed by a folded leucine-rich repeat (LRR) domain. Sds22 is conserved from yeast to human. In *S. cerevisiae*, Sds22 is an essential protein important for Glc7 nuclear targeting [41,42]. The PP1c region required for Sds22 interaction involves a relatively large number of amino acids (26 residues) and they are quite conserved in PPZ phosphatases (73% identity, 88% conservation), except those more N-terminally located (379, 380, 391 and 395 in ScPpz1). The equivalent residues in PP1c are mostly involved in the interaction with the first LRR [40]. Interestingly, Sds22 was found to physically interact with Ppz1 in diverse large-scale screenings [43,44,45]. However, it is unknown whether Sds22 could affect Ppz1 activity and function. The same pattern of conservation can be found in the Ppq1 phosphatase, for which interaction with Sds22 was also described [44]. These observations suggest that the first LRR of Sds22 could not be involved in the interaction with PPZ proteins.

Conservation of other recognized motifs in mammalian PP1c, such as the MyPhone (myosin phosphatase N-terminal element) commonly located N-terminally to the RVxF motif, is relatively high in fungal PP1. However, the conservation is only partial in the case of PPZs in Saccharomycetales and even minimal in other fungi (Appendix A). The ΦΦ motif appears strongly conserved in all fungal PP1, but not in the PPZ enzymes. Finally, the NIPP1-helix interaction pocket is well conserved in fungal PP1, but clearly less so in PPZ. It is worth noting that fungi do not seem to encode homologs of mammalian NIPP-1 proteins. In conclusion, the capacity to interact with (and perhaps be regulated by) a subset of PP1 regulatory subunits could be conserved in PPZ among most fungal species.

### 2.3. The C-Terminal Extra Alpha-Helix of CaPpz1 Is Likely Present in Most Ppz Enzymes

Chen and coworkers [32] presented evidence that CaPpz1 contains a C-terminal extra α-helix that is not present in mammalian PP1c(α). This feature is related to another characteristic difference located at the N-terminal region of the phosphatase. In CaPpz1, the N-terminal helix A’ extends an extra turn in comparison with mammalian PP1c(α), and the conformation of the loop connecting helix A’ and A (L1) is different in both phosphatases. These authors attributed a primary role for this difference to the fact that Y144 in PP1c(α) helix C is substituted by a smaller residue (C309 in CaPpz1). Remarkably, the equivalent position in ScGlc7 (Y143) is conserved in most fungal PP1 enzymes (in the few exceptions, F, an equally bulky and hydrophobic residue, is present). This same position in Ppz is also a C in all Saccharomycetales, in most Pezyzomycetes; it is also common in other clades, and, when not present, it is usually substituted by a small side chain residue (T or A). Since the authors proposed that this change results in the widening of the pocket defined by helix A’, L1, and helix B, it can be concluded that such a structural feature would be shared by virtually any fungal Ppz enzyme.

According to Chen and coworkers, the mentioned widened pocket can interact with the C-terminal end of CaPpz1 and induces its restructuring into an extra α-helix that is not present in mammalian PP1c(α). The hydrophobic CaPpz1 residues L464, L469, V472, and M473 are the key to such interactions. They are very different in PP1c(α) (and in fungal PP1 enzymes, see Appendix A). These positions are fully conserved in ScPpz1 and correspond to L654, L659, V662 and M663. As observed in Figure 5, except for V662, these positions are strongly conserved in all fungi (and when different, they are conservative changes), except for the “Monokaryotic” ones. ScPpz1 Val662 is largely conserved in other Saccharomycetales (except for *Yarrowia lipolytica*, where it is an E). Interestingly, this position is mostly occupied by non-hydrophobic residues (mainly H and E) in the rest of the fungi. Therefore, it seems likely that the extra C-terminal α-helix described for CaPpz1 will be present in most (if not all) Saccharomycetales (and possibly in most fungi), but not in “Monokaryotic” species.

### 2.4. Chemical Cross-Link of Co-Purified Ppz1 and Hal3 from S. cerevisiae Points to Possible Relevant Ppz-Specific Conserved Residues

The interaction between Ppz1 and Hal3 can occur in vivo when both proteins are simultaneously expressed in *E. coli* [39]. We took advantage of this fact and co-expressed an N-terminally 6xHis-tagged version of the catalytic domain of Ppz1 (Ppz1^Cter^) and the native full-length Hal3 protein by using a pDuet-based vector. His-tagged Ppz1 was then purified by metal affinity and, as shown in Figure 6A, Hal3 was also recovered with the phosphatase in a roughly 1:1 ratio. The mixture was subjected to chemical cross-linking with BS3 (bis(sulfosuccinimidyl)suberate), which react covalently with lysine residues, and the products analyzed by SDS-PAGE. As shown in Figure 6B, treatment with BS3 led to the formation of very high molecular mass structures that barely entered the resolving gel. These bands were excised and processed for LC-MS/MS analysis as described in Section 4.

Roughly 40% of the lysines identified by MS correspond to Hal3–Hal3 links, which is not surprising if we consider the ability of this protein to trimerize in vitro [22,46]. A number of cross-linked lysine residues corresponding to interprotein Ppz1–Hal3 cross-links were observed (Table 2), and Figure 6C displays these as a bidimensional map. It is worth noting that whereas in the case of Ppz1^Cter^ the 16 cross-linked reactive lysines are widely distributed along the polypeptide; in the case of Hal3, the links are mostly restricted to the N-terminal half and the first third of the conserved PD domain (the part that shares similarity with known PPCDC enzymes). This is interesting because previous work reported that the N-terminal extension of Hal3 is, by itself, unable to interact in vitro with Ppz1 (neither with the full-length protein nor with its catalytic domain), but it is relevant for the inhibitory function [22]. This suggests that initial interactions could involve residues located within the PD region that would be reinforced by further interactions with the N-terminal section of Hal3 (which is predicted to be highly disordered). In this regard, it is worth noting that K315, 316 and 323 within this region present consistent interactions with diverse regions of Ppz1 (Figure 6C). This scenario would agree with the notion that the N-terminal extension of Hal3 contributes to its function as a regulator of Ppz1.

Although Hal3 does contain an “RVxF”-like motif (^263^KLHVLF^268^), it is known that such a motif is not relevant for interaction with Ppz1 [47]. Although it has been described that inhibition of Ppz1 by Hal3 could happen by occlusion of the catalytic site, similar to the way inhibitor-2 inhibits PP1c, the structural features defining the interaction between Hal3 and Ppz1 are, in fact, still elusive. Our data show that Ppz1 K584 cross-links with Hal3 K90, a conserved residue in the N-terminal extension of Saccharomycetaceae (Figure 6C). Note that when K584 is mapped in our Ppz1 model, it lies very close to D566 and D615 (Figure 6D). These two residues align with Glc7 K210 and K259 and, therefore, impose a drastic change in the charge of this specific region surface. Such major changes differentiate ScPpz1 from Glc7, and they are strongly conserved in all fungi examined. It is known that Hal3 does not inhibit [30] and barely interacts with Glc7 (our own data). In contrast, the Hal3 protein from *S. cerevisiae* binds and acts as a potent inhibitor of all Ppz phosphatases (even from very distant fungi) so far examined [14,15,46,48], suggesting that all Ppz phosphatases retain key elements for interaction with ScHal3. These observations point to ScPpz1 D566 and D615 residues as possible structural elements involved in the interaction with Hal3, likely by establishing electrostatic interactions. Ppz1 K589 is relatively close to K584 and shows some interactions with Hal3 K90 too. Such interactions could also be of interest, as they lie near a cluster of three residues that appear systematically different in PP1 and PPZs (Appendix A). While the first change is rather conservative (Ppz1 V520 vs. Glc7 I164), the two next residues are negative charged in PP1 (usually D in position 165 and E in position 166) but they are always neutral in PPZs (A521 and G522 in Ppz1).

### 2.5. Mutagenesis Analysis Reveals a Key Role of Conserved D615 in Ppz1 Regulation and Function

To test the functional role of D566 and D615 in Ppz1, we changed these amino acids to K, the residues always present at the equivalent positions in PP1c. The modified phosphatases were expressed in *S. cerevisiae* from its own promoter in a centrometric plasmid to carry out in vivo functional studies. As shown in Figure 7A, immunoblot analysis of cell extracts showed that all versions of Ppz1 were expressed in yeast at similar levels. It has been known for many years that the mutation of *PPZ1* increases tolerance to lithium cations [37]. As shown in Figure 7B (left panel), reintroduction of the native phosphatase normalizes lithium tolerance. In comparison with the native phosphatase, cells expressing the D566K variant were slightly more lithium tolerant, whereas those expressing the D615K version showed a slight decrease in tolerance. The strain expressing the double mutation displayed an intermediate phenotype. This could be compatible with the D566K mutation negatively affecting Ppz1 function and the D615K change having a positive effect, perhaps due to deregulation of the enzyme. To further test this possibility, we introduced native Ppz1 and its variants in Slt2-deficient cells (strain JC10). Cells lacking the Slt2 MAP kinase are very sensitive to caffeine, and this sensitivity is attenuated by a moderate increase in Ppz1 activity [9,39], as well as by the deletion of Hal3 [20]. As shown in Figure 7B (right panel), the expression of native Ppz1 improved the growth of *slt2*Δ cells in the presence of 2 mM caffeine. An effect of similar potency was observed for the D566K and D566K D615K versions. In contrast, cells expressing the D615K variant displayed a more vigorous growth, approaching that of *slt2Δ hal3Δ* cells (strain CCV186). Thus, our results suggest that the D615K mutation renders Ppz1 abnormally active, whereas that of D566 to K may have the opposite effect.

To obtain further insight into the effect of the mutations introduced in Ppz1, the three variants plus the native enzyme were expressed and purified from *E. coli* as GST-fusion proteins by mean of glutathione beads. The immobilized phosphatases were then used as an affinity system to pull down HA-tagged Hal3 from yeast extracts prepared from strain IM021, and the material retained by the beads analyzed by SDS-PAGE and immunoblot against the HA epitope. As can be observed in Figure 7C, none of the mutations affected the ability of Ppz1 to interact with Hal3, indicating that the mutated residues were either not relevant for the interaction or that the change was insufficient to block the interaction between the regulatory subunit and the phosphatase. We then removed the GST moiety from the Ppz1 versions by protease treatment and used the enzyme preparations for in vitro testing of its capacity to be inhibited by Hal3. As shown in Figure 7D, the D566K version of Ppz1 was inhibited by Hal3 as potently as the native form of the phosphatase. In contrast, the D615K version was largely refractory to the presence of the inhibitor. Notably, the double mutant displayed an intermediate behavior. This result indicates that D615 could be a key structural element mediating Hal3 inhibition and would agree with the increase in function deduced from the phenotypic results (Figure 7B). In contrast, the D566K version, which is inhibited in vitro by Hal3 as much as the native phosphatase, showed some decrease in function in vivo. We determined the specific activity of the different variants expressed in *E. coli* and found that, compared to the native recombinant Ppz1, the D566K version systematically appeared slightly less active (81.5% ± 8.2, n = 3), whereas the D615K variant was indistinguishable from the native enzyme (99.3 ± 1.3, n = 3). Therefore, a lesser catalytic activity of the D566K version may explain the in vivo results. The intermediate behavior in vivo of the double mutated version may be explained by a compensatory effect of both mutations.

The observation that the D615K mutation abolishes the ability of Hal3 to inhibit Ppz1 suggests that electrostatic interactions may play a role in the inhibitory mechanism. The scenario would fit with the early proposal that the in vivo interaction between Ppz1 and Hal3 is dependent on the intracellular pH [49]. It must be noted that previous work [39] revealed two acidic residues (E575 and E630) whose mutation to G resulted in a marked decrease in the ability of Hal3 to inhibit Ppz1 without significant change in the interaction. This could imply that the interaction between Ppz1 and Hal3 involves diverse structural determinants that are redundant, so elimination of one of them has little or no effect in the interaction. These residues are also near the catalytic site (in purple in Figure 6D). However, in contrast to E615, E575 and E630 are conserved not only in Glc7 (D219 and E274, respectively), but also in the vast majority of fungal species analyzed in this work (see Appendix A). All these results indicate that the conserved D615 in Ppz1 is an important structural determinant that contributes to the specificity of the regulation of Ppz and PP1c enzymes in fungi.

## 3. Conclusions

We analyzed the sequences of PP1 and the catalytic domains of PPZ proteins of multiple species representative of all three main fungi clades. Our data indicate that Ppz1 proteins are more diverse than PP1 proteins, and this is much more evident if we exclude the Sds21 family of PP1. Characteristic residues of each type of phosphatases mainly mapped on the surface of the modeled proteins and, in some cases, involved a change in the net charge. This fact could explain, at least in part, the specificity of some regulatory subunits for binding to PP1 or PPZ. Our data show, for example, that several of the motifs required for the binding of PP1 to its regulatory subunits are not present or only partially conserved in PPZ1 proteins. We also propose specific structural features, such as an extra C-terminal α-helix, to be differential landmarks between Ppz and PP1c enzymes. Finally, comparative analysis of dozens of PPZ and fungal PP1 proteins allowed the identification of a strongly conserved (and differential) residue (D615 in ScPpz1) that appears crucial for inhibition by Hal3.

## 4. Materials and Methods

### 4.1. Yeast Strains

Yeast cells were incubated in YP medium (1% yeast extract, 2% peptone) or in synthetic medium (SC) lacking uracil [50], supplemented with glucose at 2% as a carbon source, at 28 °C. Plates contained 2% agar. The *ppz1*Δ mutant is a kanMX deletant derived from strain BY4741 (*MAT***a** *his3*∆1 *leu2*∆ *met15*∆ *ura3*∆) [51]. Strains JC010 (*slt2*Δ::*LEU2*) and CCV186 (*slt2*Δ::*LEU2 hal3*Δ::*KanMX*) are JA100 (*MAT***a** *ura3*-52 *leu2*-3,112 *trp1*-1 *his4 can*-1r) derivatives and are reported in Refs. [11] and [48], respectively. Strain IM021 (*ppz1*Δ::*KanMx4 hal3*Δ::*LEU2*) is reported in [47].

### 4.2. Plasmid Construction

*Escherichia coli* DH5α cells, employed as plasmid DNA host, were grown at 37 °C in LB medium supplemented with 50 μg/mL ampicillin when carrying plasmids. Transformations of *S. cerevisiae* and *E. coli*, and standard recombinant DNA techniques were performed as described [52]. Mutation of Ppz1 D566 to K was done as follows. Two partially overlapping fragments were PCR amplified from vector pRS316-Ppz1 [39] and oligonucleotides Ppz1_5′_seq_cat/3_Ppz1_D566K (fragment 1), and M13rev/5Ppz1_D566K (fragment 2). Both fragments were mixed and subjected to a second round of amplification with primers Ppz1_5′_seq_cat/M13rev. The amplification fragment was digested with PacI and HindIII and used to replace the equivalent fragment in pRS316-Ppz1. The mutation of D615 to K was done in the same way but using primers 5_Ppz1_D615K and 3_Ppz1_D615K. The double mutant was constructed by introducing the D615K change in the pRS316-Ppz1^D566K^ construct. The mutated versions were introduced in pGEX6P-1 for expression in *E. coli* by digestion of the pRS-based construct with Kpn2I and Bsp1407I and replacement of the same region in pGEX6P-1-Ppz1 [47]. Plasmid pGEX6P-Hal3 was used to express Hal3 fused to GST in *E. coli* [30].

The sequence of oligonucleotides mentioned in this work can be found in Appendix A.

### 4.3. Bioinformatics Analyses

Using the whole-proteome-based fungi phylogeny as a reference [35], we selected 57 species representative of the Ascomycota (35), Basidiomycota (15) and Monokaryotic fungi (7) and identified their Ppz and PP1 proteins. For this purpose, we used the Ppz1 and Glc7 protein sequences from *S. cerevisiae* to perform BLAST [53] searches in the selected species at the NCBI server (https://blast.ncbi.nlm.nih.gov/Blast.cgi, accessed on 20 April 2021). Top hits were submitted to BLAST search against the *S. cerevisiae* proteome at the SGD webpage [54] (https://www.yeastgenome.org/blast-sgd, accessed on 20 April 2021) and classified as PP1 or Ppz1 accordingly.

Because the conserved catalytic domain of Ppz proteins does not start in the first residues as it does in PP1 proteins, we removed the non-catalytic N-terminal residues of Ppz for further sequence alignment and comparison with PP1. To this end, Ppz proteins were scanned for the localization of the SM000156 Smart domain [55], which begins at residue 385 in *S. cerevisiae* Ppz1. However, it was clear that the similarity between PP1 and Ppz extends further toward the N-terminus. Therefore, we found that the protein phosphatase domain PTHR11668, predicted by the Panther evolutionary classification tool [56], which initiates at residue 359 in ScPpz1, was more suitable to our purposes. Unless otherwise stated, the numbering of residues relates to ScGlc7 and ScPpz1 sequences, respectively.

The Clustal Omega tool available at the EMBL-EBI server [57] was used to create the sequence alignments. Shading of residues according to their conservation across species was done with pyBoxshade, a desktop updated version of the Boxshade WEB interface (https://embnet.vital-it.ch/software/BOX_form.html, downloaded 3 May 2021).

Proteomes were obtained from Uniprot (https://www.uniprot.org/, accessed on 20 May 2021), and the number of Ser and Thr for each species was obtained with a script developed in Visual Basic. Theoretical pI were calculated using the ProtParam tool available at the Expasy server [58].

Models of *S. cerevisiae* Glc7 and Ppz1 were generated at the Swiss Model Server (https://swissmodel.expasy.org/, accessed on 23 December 2020) [59]. The numbering corresponds to the full-length proteins. Sequence logos were generated with the Seq2Logo generator (https://services.healthtech.dtu.dk/, accessed on 10 June 2021), as described in [60]. The Shannon logo type was selected; clustering method was set to none, and “Weight on prior” was set to 0. The coloring scheme was the default, except that Phe was denoted in yellow.

### 4.4. Chemical Cross-Linking of the Ppz1 Catalytic Domain and Hal3

Co-purification of the Ppz1-Cter/Hal3 complex was accomplished upon co-expression of both proteins in *E. coli* BL21 DE3 RIL cells transformed with a pET-Duet1-based plasmid bearing the C-terminal half (Δ1–344) of Ppz1 (N-terminally fused to a 6xHis tag) and the entire Hal3, as described in [23].

Bacterial cultures were grown in a shaker at 37 °C to an OD_600_ of 0.6–0.8. Recombinant protein expression was initiated by addition of isopropyl-ß-D-thiogalactopyranoside (IPTG) at 0.1 mM and continued for 16 h at 21 °C. Cultures were centrifuged at 6500× *g* for 20 min at 4 °C, and pellets were resuspended in lysis buffer (50 mM NaH_2_PO_4_ pH 7.5, 150 mM NaCl, 20 mM Imidazole, 0.1% Triton X-100) containing Complete EDTA-free protease inhibitor cocktail (Roche, Sant Cugat del Vallès, Spain) plus 2 mM PMSF (approximately 3 mL of buffer per gram of pellet). Samples were placed on ice and sonicated using a Vibra-cell (Sonics and Materials Inc.) equipment for 5–6 min (10 s on, 10 s off) at 30% amplitude. Samples were then centrifuged at 4 °C (12,400× *g* for 30 min) to remove cellular debris. The soluble fraction was incubated with Ni-NTA Agarose beads (Qiagen^®^, Hilden, Germany) following the manufacturer’s specifications and loaded in a 10 mL column. The mixture was washed with about 5 bed volumes of lysis buffer (without Triton X-100) until the eluate was free of proteins (monitored by the Bradford’s reagent). The 6x-His tagged Ppz1-Cter/Hal3 complex was recovered by incubating the beads for 10 min with 1.5 bed volume of elution buffer (washing buffer containing 150 mM imidazole). Finally, imidazole was eliminated by exchanging the elution buffer with cross-linking buffer (50 mM NaH_2_P0_4_ pH 7.5, 150 mM NaCl) using an Amicon^®^ Ultra Centrifugal filter (size 30 kDa, Millipore, Burlington, MA, USA). The amounts of the recovered recombinant proteins were determined by SDS-PAGE followed by Coomassie-blue staining and scanning of the gels with an Epson Perfection V500 Photo Scanner. Quantification was done with the Gel Analyzer software (version 2010a) and different amounts of bovine serum albumin (BSA) as standards.

The purified Ppz1C-ter/Hal3 protein complex was adjusted to 0.2–1 μg/μL with cross-linking buffer. Forty micrograms of the complex was used in each experiment. The proteins were incubated for 1 h at room temperature with 5, 10 and 20 weight excesses of BS3 (bis(sulfosuccinimidyl)suberate) (Covachem LLC, Loves Park, IL, USA). Cross linking was then quenched by addition of NH_4_CO_3_ at 100 mM. Then, samples were analyzed by SDS-PAGE (6% polyacrylamide gels) and stained. The relevant bands were excised from the gel into individual low-binding polypropylene tubes. In-gel tryptic digestion was performed essentially as described [61]. Each sample was desalted using a Rainin pipette tip packed with Poros R2 reversed phase material (Applied Biosystems, Foster City, CA, USA).

### 4.5. LC-MS/MS Analysis of Cross-Linked Proteins

The digested peptides were dissolved in 0.1% TFA, and approximately 1μg peptides were analyzed using an EASY-nanoLC 1000 system (Thermo Scientific, Dreieich, Germany) coupled to a Q-Exactive HF mass spectrometer. The peptides were loaded onto a 2.5 cm in-house packed Reprosil-Pur 120 C18-AQ (5 μm: Dr. Maisch GmbH, Germany) precolumn with an internal diameter of 100 μm and eluted directly onto a 19 cm in-house packed Reprosil-Pur C18-AQ column (3 μm; Dr. Maisch GmbH, Ammerbuch, Germany) with an internal diameter of 75 μm. A 104 min HPLC gradient with a flow rate of 250 nL/min was used, with an increasing concentration of Solvent B (95% ACN, 0.1% FA) in the following increments: 1–3% for 3 min, 3–25% for 80 min, 25–45% for 10 min, 45–100% for 3 min and 100% for the final 8 min. Solvent A was 0.1% FA.

The mass spectrometer was operated in data-dependent acquisition mode, and each analysis started with an MS1 scan detected in the Orbitrap (resolution: 120.000, scan range: 300–1600 *m*/*z*, AGC target: 1e6, maximum injection time: 100 ms, RF lens: 30%). For each MS1 spectrum, the 12 most intense peaks were selected for MS2. Monoisotopic peak determination was set to peptide, and dynamic exclusion was switched on (exclude after 1 time, exclusion duration: 7 s, mass tolerance: ±10ppm, exclude isotopes). The selected MS2 precursors were isolated in the quadrupole at 1.2 Th (*m*/*z*) and fragmented using HCD at a collision energy of 32%. MS2 fragments were detected in the Orbitrap (resolution: 30.000, first mass: 110 *m*/*z*, AGC target: 1e4).

Raw data files were converted to mgf format by Rawconverter 1.2.0.1 (Scripps Research Institute, La Jolla, CA, USA). The data were further filtered using an “MGF filter” (www.massai.dk, Stenstrup, Denmark) using the following settings: Only scans with charge states between +3 and +9 were retained to maximize the number of scans containing highly charged cross-links. Only scans containing signature peaks for tryptic digests (y1 ions) and at least one signature peak from the cross-linker itself were kept. The sequence of ScPpz1 (P26570) and Hal3 (P36024) were acquired from the UniProt database (www.UniProt.org, accessed on 1 April 2021). The sequences were used for examining BS3 crosslinks by the MassAI search engine using the parameters: 10 ppm MS accuracy, 0.10 Da MS/MS accuracy, trypsin for the enzyme, four allowed missed cleavages, carbamidomethylation of cysteine as fixed modification and oxidation of methionine, BS3-water and BS3-ammonia dead-end of lysine residues and the protein N-terminus as variable modifications. Cross links were manually validated, and cross-link spectra were only approved if b- and/or y-ions from both involved peptides were present in the spectrum. Representative MS/MS scans are shown in Appendix A.

### 4.6. Preparation of Yeast Extracts and Immunoblotting

Protein extracts for Ppz1 detection were prepared from 10 mL of cultures and subjected to SDS-PAGE and immunoblot as described [62,63]. Ppz1 was detected with polyclonal anti-Ppz1 antibodies (1:250 dilution) followed by secondary anti-rabbit IgG-horseradish peroxidase antibodies (GE Healthcare, 1:20,000 dilution) and the ECL Prime Western blotting detection kit. For detection of HA-tagged Hal3, yeast extracts containing HA-tagged Hal3 were prepared from strain IM021 (*ppz1*Δ *hal3*Δ) as described [15], except that 50 mL cultures were grown and 500 μL of lysis buffer was used for cell disruption. After cell lysis, 125 μL of the lysis buffer was added to increase the extraction volume. Immunodetection of HA-Hal3 was done as in [15], except that the anti-HA antibody was from Biolegend.

### 4.7. Hal3 Pull down Experiments

GST-Ppz1, Ppz1^D566K^, Ppz1^D615K^, and Ppz1^D566K, D615K^ were expressed in BL21 (DE3) RIL *E. coli* cells and purified as reported [22] with the modifications described in [14]. Quantification of the expressed phosphatase versions was done as in [15]. The interaction between the Ppz phosphatases and Hal3 was determined by in vitro binding assays performed as previously described [14,22] with minor modifications. Thus, aliquots of glutathione agarose beads containing 8 μg of the different Ppz1 versions were incubated with protein extracts (0.6 mg of proteins) prepared from strain IM021 (*ppz1*Δ *hal3*Δ). The beads were finally resuspended in 120 μL of 2x_sample buffer and, after boiling for 5 min, 40 μL of each sample was analyzed by SDS-PAGE and HA-Hal3 immunodetection performed as described above.

### 4.8. Ppz1 Inhibition Assays

For Ppz1 inhibition assays, all recombinant proteins were expressed in *E. coli* BL21(DE3) RIL cells as described in [39,62], except that Terrific Broth was used instead of LB medium. Removal of the GST moiety from recombinant Ppz1 versions and Hal3 was done essentially as described in [62]. Phosphatase activity was determined using *p*-nitrophenylphosphate as substrate and five picomoles of the different phosphatases.

## Figures and Tables

**Figure 1 ijms-23-01327-f001:**
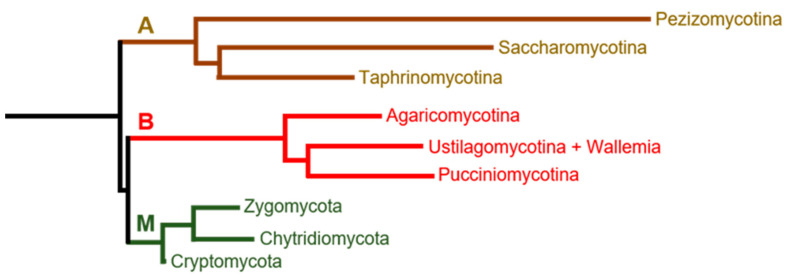
Simplified phylogenetical tree of the different clades covered in this study. Branches labeled as A, B and M denote the Ascomycota, Basidiomycota and “Monokarya” fungi clades. The figure was constructed based on data from reference [35].

**Figure 2 ijms-23-01327-f002:**
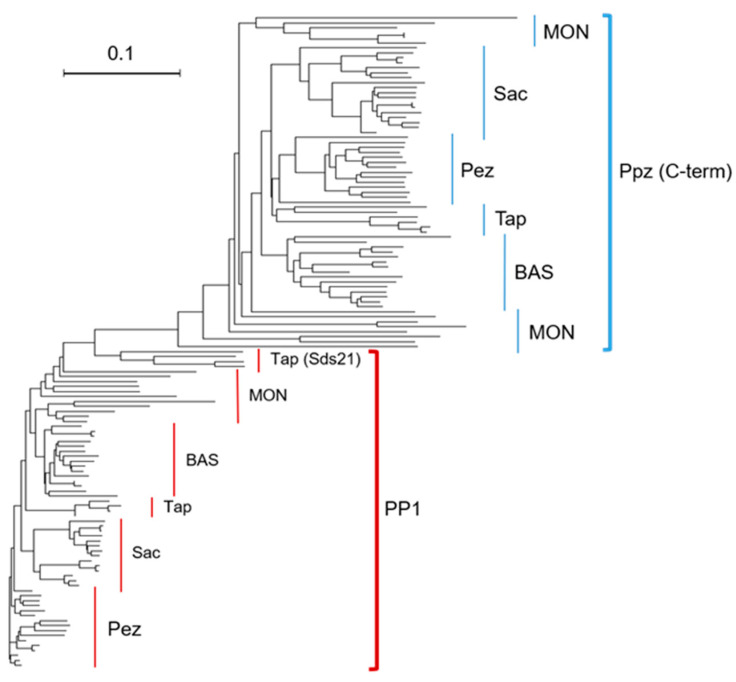
PP1 enzymes are more conserved than PPZ across fungi. Phylogenetic tree containing the sequences of PP1 and the catalytic regions of PPZ proteins. Phylogenetic tree data generated from Clustal Omega alignments was visualized using the Dendroscope software (version 3.7.3) as rectangular phylogram [36]. Pez: Pezizomycotina. Sac: Saccharomycotina. Tap: Taphrinomycotina. BAS: Basidiomycota. MON: “Monokaryotic” fungi.

**Figure 3 ijms-23-01327-f003:**
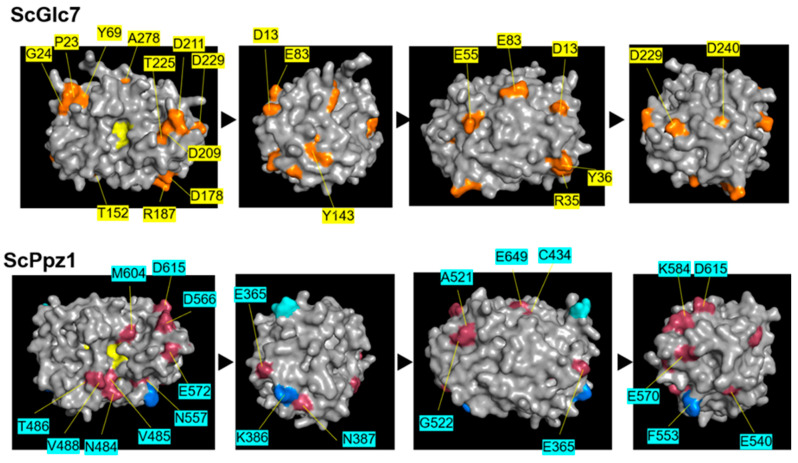
Mapping of relevant surface residues that differ in Ppz1 and Glc7. The upper panel identifies class B residues on the surface of a Glc7 model (orange). In the lower panel, residues in class A (raspberry) or C (blue) are mapped on the surface of the *S. cerevisiae* Ppz1 model for its catalytic region. Residues forming the catalytic site are shown in yellow. The transition between images corresponds to a rotation (anticlockwise) of 90⁰ on the *Y*-axis.

**Figure 4 ijms-23-01327-f004:**
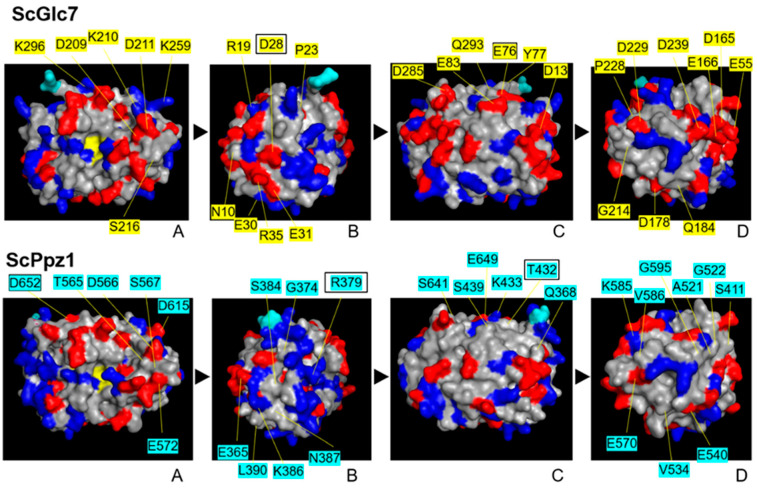
Distribution of changes affecting charged residues on the surface of both Glc7 (**upper panel**) and the catalytic region of Ppz1 (**lower panel**). Mapped residues represent potential alterations in surface charges in PP1 or PPZ enzymes. Acidic residues are in red and basic ones in dark blue. Boxed residue names are changes present in *S. cerevisiae* Glc7 or Ppz1 but not necessarily in most other species. The C-terminal modeled residue of Glc7 and Ppz1 are depicted in cyan to facilitate identification. A to D denotes the positions after 90⁰ anticlockwise rotation on the *Y*-axis.

**Figure 5 ijms-23-01327-f005:**
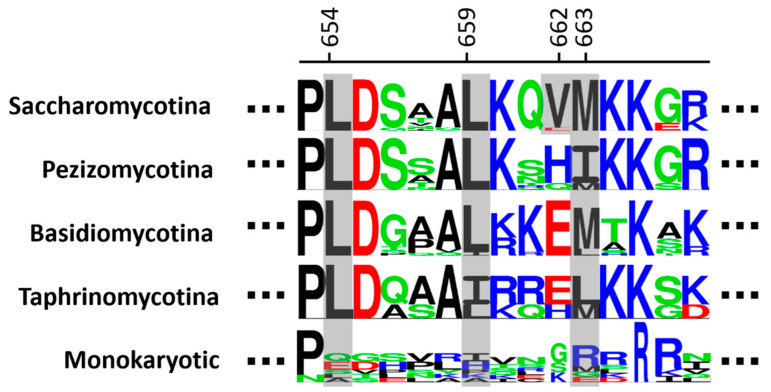
Conservation of the C-terminal extra alpha-helix in PPZ proteins. The abundance of specific residues at the C-terminus of PPZ proteins is depicted as logos grouping the different clades indicated on the left. Numbering corresponds to the *S. cerevisiae* Ppz1 enzyme. The residues discussed in the text are shaded in grey.

**Figure 6 ijms-23-01327-f006:**
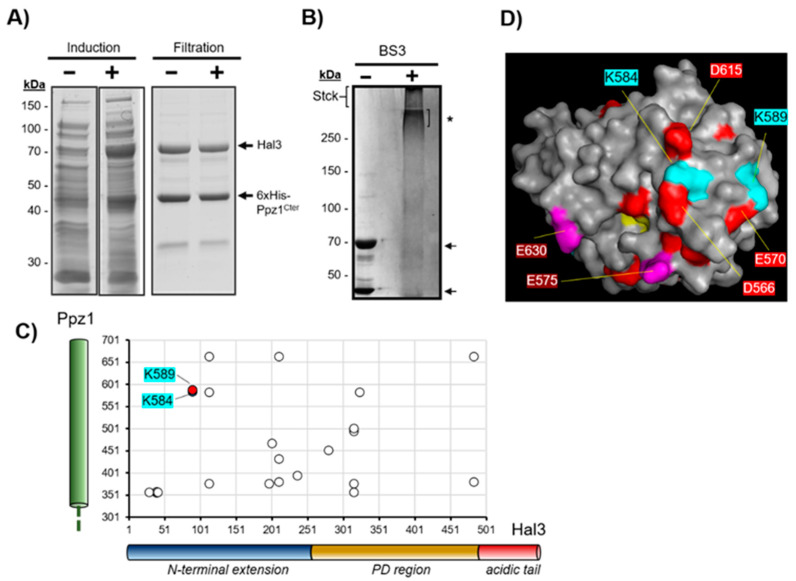
Chemical cross-link between the catalytic domain of Ppz1 and Hal3. (**A**) The catalytic domain of Ppz1 tagged with a 6xHis sequence at the N-terminus (6xHisPpz1^Cter^) was co-expressed with full-length Hal3. Upon metal-affinity purification and change of the buffer by filtration, samples were electrophoresed and stained with Coomassie blue. (**B**) Samples of the Ppz1^Cter^ plus Hal3 mixture were incubated with BS3 (10× excess) as described and electrophoresed in 6% polyacrylamide gels prior staining. Arrows indicate the position of the monomeric proteins and the asterisk (*) the high molecular mass complex excised for MS determination. “Stck” denotes the position of the stacking gel (not removed prior imaging). (**C**) Bidimensional map showing the relative positions of cross-linked K in Ppz1^Cter^ and Hal3. Relevant K residues discussed in the text are highlighted in color. (**D**) Mapping of K584 and K589 in a surface model of Ppz1. Residues that differ between Ppz1 and Glc7 are in red, and catalytic residues are in yellow. Purple residues correspond to amino acids previously identified as relevant for Ppz1 inhibition [39]. See main text for discussion.

**Figure 7 ijms-23-01327-f007:**
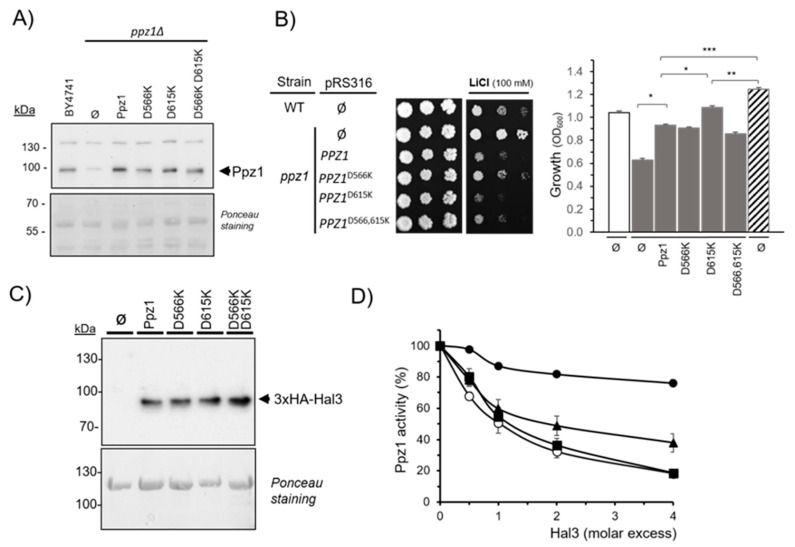
Functional characterization of the D566K and D615K mutations in Ppz1. (**A**) Forty micrograms of protein extracts from the wild-type strain BY4741 and *ppz1*Δ cells bearing the different variants of Ppz1 (Ø, empty plasmid) were subjected to SDS-PAGE (10% polyacrylamide gels) and were transferred to membranes. The Ppz1 versions were detected with a polyclonal anti-GST-Ppz1 antibody. (**B**) Left panel. Wild-type and *ppz1*Δ cells were transformed with the diverse versions of Ppz1 and cultures spotted (OD_660_ = 0.05 and at 1/5 serial dilution) on synthetic media (lacking uracil) plates with or without 100 mM LiCl. Pictures were taken after six days. Ø, empty plasmid. Right panel. Strains JA100 (wild type, open bar), JC10 (*slt2*Δ, grey bars) and CCV186 (*slt2*Δ *hal3*Δ, striped bars) were transformed with the indicated plasmids and growth in liquid culture in the presence of 2 mM caffeine (initial OD_600_ = 0.004) as in [33]. Growth was monitored every 30 min in a BioScreen C apparatus (Thermo Fisher Sci., Waltham, MA, USA) and a representative result after 24 h is shown. Data are mean ± SEM from three independent clones determined at least by duplicate. *, *p* < 0.01; **, *p* < 0.001; ***, *p* < 0.0001 (**C**) Interaction of the different versions of Ppz1 with Hal3. Equal amounts of the indicated versions of GST-tagged phosphatases were immobilized on glutathione beads and incubated with protein extracts of strain IM021 (*ppz1*Δ *hal3*Δ) expressing a HA-tagged version of Hal3. Beads were washed and processed for SDS-PAGE (8% polyacrylamide gels) and immunoblotting using anti-HA antibodies as described in Materials and Methods. Ponceau staining shows the amount of GST-Ppz1 in the assay. (**D**) The different versions of Ppz1 (○, native; ■, D566K; ●, D615K; ▲, D566K D615K) were incubated with increasing amounts of bacterially expressed Hal3. Data are represented as the percentage of phosphatase activity in the absence of inhibitors and correspond to the mean ± SEM from at least three independent determinations using two different preparations of each protein.

**Table 1 ijms-23-01327-t001:** Distribution of selected organisms and their analyzed sequences among different fungal clades.

	Phylum	Species	PP1	Ppz
Ascomycota	Saccharomycotina	15	15	18
	Pezizomycotina	13	13	13
	Taphrinomycotina	7	11	7
Basidiomycota	Agaricomycotina	10	11	10
	Wallemiomycetes Ustilagomycotina Pucciniomycotina	5	5	5
“Monokaryotic”	Zygomycota Chytridiomycota Cryptomycota	7	9	14
	Total number	57	64	67

**Table 2 ijms-23-01327-t002:** Relevant lysine residues in the catalytic domain of ScPpz1 that cross-link with Hal3 residues.

Ppz1	Hal3
356/359	39
358	29, 41, 315
378	113, 197, 315, 316
381	210, 483
396	236
433	210
453	280
468	201
496/502	315
584	90, 113, 323
589	90
664	113, 210, 483

Note: Based on the MS/MS fragmentation spectra it has not been possible to differentiate between K356/K359 and between K496/K502.

## Data Availability

Data supporting the reported results are presented in this manuscript and its Appendix A.

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
