# Peer review of "Comparative Analysis of Type 1 and Type Z Protein Phosphatases Reveals D615 as a Key Residue for Ppz1 Regulation"

_ijms, 2022, doi:10.3390/ijms23031327_

Round 1
Reviewer 1 Report
Interesting and well written article
Author Response
We appreciate very much the positive input from the referee.
Reviewer 2 Report
Summary:
In the present paper, the authors present their research results that was carried out in order to demonstrate that in fungi the most important determinant that contribute to the differential regulation of type 1 and type Z protein phosphatases (Ppz and PP1c) is the conserved D615.
The hypothesis is interesting, the study is original and the study design is appropriate. The methods are clearly described and the manuscript is well structured. The authors contribute to this field of research with new findings. Nevertheless, some minor revisions are recommended before publication.
Observations:
- Please, add a Conclusion section
- Please, revise and correct the style of number 21, 52, 54 and 60 of references.

Author Response
Observations:
1.- Please, add a Conclusion section
R: Following the advice of the referee, we have included a Conclusions section as a new section 3.
2.-Please, revise and correct the style of number 21, 52, 54 and 60 of references.
R: References have been revised and corrected according the latest version of IJMS style available from Mendeley.
Reviewer 3 Report
This manuscript presents a study of regulation of fungal protein phosphatase PPZ. The study includes multidisciplinary approach including bioinformatics analysis, structural modeling, MS analysis, in vitro protein-protein interaction, enzyme activity, and growth of mutant strains of yeast. The conclusion is clear. Methods were described in detail. However, supplementary data were not provided for this review. Thus, it is very difficult to judge the conclusion/discussion of the manuscript. This reviewer cannot agree with some of the conclusions drawn from the experimental results. There are some proof reading errors in the text. These should be fixed for better understanding and clarity. Followings are specific comments.
Description of yeast genotype: The genotype of yeasts are described with or without “Δ.” For example, strain IM021 (ppz1Δ hal3Δ) or strain IM021 (ppz1 hal3) in the text and ppz1Δ and ppz1 in Figure 7. Are there any specific reason for this usage? If not please use one nomenclature. It is confusing.
Amino acid code: The one-letter code and three-letter code were used in the text randomly. This is confusing. Please use just one code.
L217-218: This statement seems premature since functional contribution is unknown (L214-215).
Figure 6D: What are pink/purple residues? Description of this is missing in the figure legend.
L338-339: Is this simply because the activity of D615K is smaller than WT and D566K?
L345-347: To this reviewer the growth of yeast expressing three different version of PPZ1 is very similar. This reviewer cannot agree with this description. In addition, there is no discussion about the double mutant. To this reviewer, the double mutant is the only one that shows difference in growth/caffeine sensitivity.
Figure 7B: the figure legend and manuscript text indicated that yeast mutant stl2 was used in the experiment. To this reviewer this mutant seems to be stl2 ppz1. If so please correct this error.
L375: What is the nature of “recombinant ScHal3?” How is was this prepared?
L390-391: As indicated, the double mutant showed difference from the two single mutant. There should be some discussion. Is this an indication of D566 involvement in regulation via Hal3?
Figure 7D: The activity is shown as relative to the activity without Hal3. What are the absolute activity of the four version of PPZ1? This should be presented. It is important to interpret data in Figure 7B.
L431-432: GST-tagged Hal3 was not described in the Results and Discussion section. Is this error?
Minor points:
L30: “:” is in red.
L413-414: the genotype of IM021 has typo.
L548: Font mismatch.
Author Response
Please, find our response to th ereferee in the attached document

Round 2
Reviewer 3 Report
I am satisfied with the responses of authors and new version of the manuscript. I found a typo in the text (L211; LLR). It should be fixed before publication.